# MProtoNet: A Case-Based Interpretable Model for Brain Tumor Classification with 3D Multi-parametric Magnetic Resonance Imaging

**Yuanyuan Wei**[1,2]                                                      AYWI@STUDENT.UBC.CA

**Roger Tam**[*2]                                                          ROGER.TAM@UBC.CA

**Xiaoying Tang**[*1]                                                      TANGXY@SUSTECH.EDU.CN

[1] *Department of Electronic and Electrical Engineering, Southern University of Science and Technology, Shenzhen, China*

[2] *School of Biomedical Engineering, The University of British Columbia, Vancouver, Canada*

**Editors:** Accepted for publication at MIDL 2023

## Abstract

Recent applications of deep convolutional neural networks in medical imaging raise concerns about their interpretability. While most explainable deep learning applications use post hoc methods (such as GradCAM) to generate feature attribution maps, there is a new type of case-based reasoning models, namely ProtoPNet and its variants, which identify prototypes during training and compare input image patches with those prototypes. We propose the first medical prototype network (MProtoNet) to extend ProtoPNet to brain tumor classification with 3D multi-parametric magnetic resonance imaging (mpMRI) data. To address different requirements between 2D natural images and 3D mpMRIs especially in terms of localizing attention regions, a new attention module with soft masking and online-CAM loss is introduced. Soft masking helps sharpen attention maps, while online-CAM loss directly utilizes image-level labels when training the attention module. MProtoNet achieves statistically significant improvements in interpretability metrics of both correctness and localization coherence (with a best activation precision of $0.713 \pm 0.058$) without human-annotated labels during training, when compared with GradCAM and several ProtoPNet variants. The source code is available at https://github.com/aywi/mprotonet.

**Keywords:** Interpretable Deep Learning, Explainable AI, Brain Tumor Classification

## 1. Introduction

With the development of architecture design and training strategy of very deep neural networks, models such as deep convolutional neural networks (CNNs) have achieved state-of-the-art performance in image classification, object detection, semantic segmentation and the respective applications in medical imaging (Zhou et al., 2021). However, recent applications of CNNs in medical imaging raise questions about their interpretability since high-stake decisions are made upon these inherently complex models (Rudin, 2019). Specifically, CNNs are known to be vulnerable when taking undesired shortcuts during training (Geirhos et al., 2020). Requirements in medical settings emphasize the evaluation of interpretability since the predictions of CNNs may not rely on intended evidence (DeGrave et al., 2021).

---

\* Corresponding authors: Dr. Xiaoying Tang; Dr. Roger Tam.

Most explainable deep learning applications, including those in medical imaging, use post hoc methods to generate feature attribution maps after training of CNNs. Representative methods include the well-known class activation mapping (CAM) (Zhou et al., 2016) and its more general variant GradCAM (Selvaraju et al., 2017). These post hoc methods nevertheless are criticized for providing localization-only and often unreliable explanations (Adebayo et al., 2018; Rudin, 2019). There are also some attempts to incorporate interpretable blocks in the design of CNNs, such as case-based and concept-based reasoning models. Compared to concept-based interpretable models (Koh et al., 2020) that require predefined concepts, case-based interpretable models (Chen et al., 2019) can identify prototypes during training and compare similarities between the input image patches and the identified prototypes, resulting in both attribution maps and case-based prototypical explanations.

**Related Work**   ProtoPNet (Chen et al., 2019) is the first case-based interpretable deep learning model. While there are many attempts to further improve the performance of ProtoPNet on multi-class natural image classification (Wang et al., 2021; Rymarczyk et al., 2021, 2022; Donnelly et al., 2022), the investigation of ProtoPNet and its variants on medical imaging applications is still relatively limited. Barnett et al. (2021) evaluates a ProtoPNet variant (IAIA-BL) on breast lesion classification with 2D digital mammography images, with additional costs of fine-grained annotations from clinicians. XProtoNet (Kim et al., 2021) is another application on chest radiography with 2D X-ray images, but there are no quantitative evaluation results on interpretability such as localization coherence. To enhance the localization performance of CNNs, online-CAM modules (Fukui et al., 2019; Ouyang et al., 2021) show their great potential by directly utilizing image-level labels during training of the attention modules. Similar methods have also been used in weakly supervised detection (Amyar et al., 2022) and segmentation (Zhou et al., 2018).

**Contributions**   Our contributions can be summarized as follows: (1) We propose the first medical prototype network (MProtoNet) to extend ProtoPNet to brain tumor classification using 3D multi-parametric magnetic resonance imaging (mpMRI) data which present additional challenges compared to both natural images (Chen et al., 2019; Donnelly et al., 2022) and 2D medical images (Barnett et al., 2021; Kim et al., 2021), especially in terms of localizing attention regions. (2) A new attention module is proposed to enhance the localization of attention regions, along with soft masking that sharpens attention maps and online-CAM loss that directly utilizes image-level labels during training. (3) Interpretability metrics of correctness and localization coherence are employed throughout all experiments to evaluate the interpretability of the proposed MProtoNet and other compared models. The source code is available at https://github.com/aywi/mprotonet.

## 2. Methods

### 2.1. Dataset

We use the multimodal brain tumor image segmentation (BraTS) (Menze et al., 2015) 2020 dataset[1] to validate our methods throughout this work. The BraTS 2020 dataset contains 369 subjects with pathologically confirmed diagnoses: 293 with high-grade glioma (HGG)

---

1. https://www.med.upenn.edu/cbica/brats2020/

and 76 with low-grade glioma (LGG). Each subject has mpMRI data of four modalities: T1-weighted, T1-weighted contrast enhancement (T1CE), T2-weighted and T2 fluid attenuated inversion recovery (FLAIR). Originally, for each subject in BraTS 2020 there are three tumor sub-regions labeled. In this work, we unify all sub-region labels as the single whole tumor (WT) region and only use this information when evaluating the interpretability metrics. The preprocessing and data augmentation details are described in Appendix A. After preprocessing, each image has a size of $128{\times}128{\times}96$ voxels ($1.5{\times}1.5{\times}1.5$ mm$^3$ resolution).

## 2.2. Architecture

The overall architecture of MProtoNet is shown in Figure 1. It consists of four layers: a feature layer, a localization layer, a prototype layer, and a classification layer.

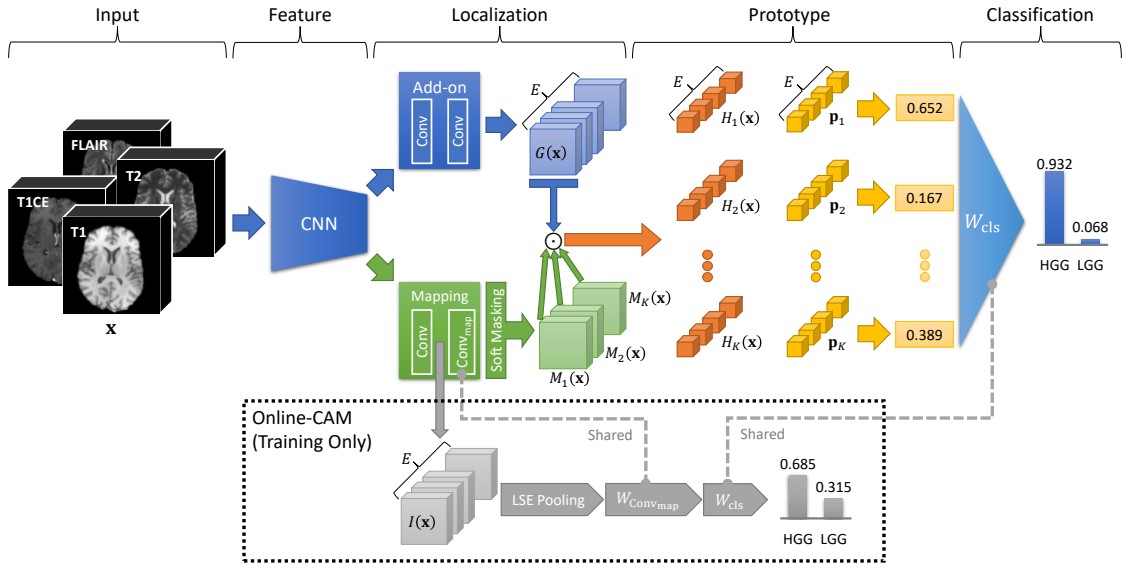

Figure 1: The overall architecture of MProtoNet. Online-CAM only occurs during training.

**Inference Process** All layers are described in detail below. At a high level, MProtoNet delivers its prediction by comparing processed features with a set of learned prototypes $\mathbf{p}_k \subseteq \mathbf{P}$ ($k \in \{1 \ldots K\}$). Given input mpMRI data $\mathbf{x}$, the feature layer extracts the embedding features $F(\mathbf{x})$. After that, $F(\mathbf{x})$ is sent to both an add-on module and a mapping module in the localization layer to get high-level features $G(\mathbf{x})$ and an attention map $M_k(\mathbf{x})$ for each prototype $\mathbf{p}_k$. The dot-product result between $G(\mathbf{x})$ and each $M_k(\mathbf{x})$ is treated as the final features $H_k(\mathbf{x})$ to be compared with each prototype $\mathbf{p}_k$. In the prototype layer, a similarity score $s(\mathbf{x}, \mathbf{p}_k)$ between each pair of $H_k(\mathbf{x})$ and $\mathbf{p}_k$ is calculated and sent to the classification layer to get the final prediction.

**Feature Layer** The feature layer is a CNN module that performs feature extraction on the input mpMRI $\mathbf{x} \in \mathbb{R}^{4 \times 128 \times 128 \times 96}$. ResNet (He et al., 2016) is chosen as the backbone for

its good performance on a wide variety of image tasks and relatively low model complexity. We directly replace 2D operations (such as convolutions) with 3D ones to make a 3D version of ResNet. Due to the need for localization at subsequent layers, we only use the layers up to the second building block of ResNet-152, resulting in a final output $F\left(\mathbf{x}\right) \in \mathbb{R}^{512 \times 16 \times 16 \times 12}$ sent to the localization layer. Because it is difficult to find a suitable pre-trained version of 3D ResNet, the feature layer is trained from scratch.

**Localization Layer** The localization layer has two branches that receive the same input $F\left(\mathbf{x}\right)$ from previous layer: an add-on module and a mapping module. The add-on module (following ProtoPNet (Chen et al., 2019)) extracts high-level embedding features $G\left(\mathbf{x}\right) \in \mathbb{R}^{E \times 16 \times 16 \times 12}$, where $E$ is the number of embedding channels. The mapping module (modified from XProtoNet (Kim et al., 2021)) outputs attention maps $M_k\left(\mathbf{x}\right) \in \mathbb{R}^{16 \times 16 \times 12}$ ($k \in \{1 \ldots K\}$) for localizing regions related to prototypes $\mathbf{p}_k$ ($k \in \{1 \ldots K\}$). The values of the attention maps $M_k\left(\mathbf{x}\right)$ are scaled into a range of $[0, 1]$. To learn more accurate and sharper attention maps and reduce irrelevant background areas, we introduce differentiable soft masking (Li et al., 2018) to sharpen the values of the attention maps $M_k^0\left(\mathbf{x}\right)$

$$M_k\left(\mathbf{x}\right) = \frac{1}{1 + \exp\left(-\omega\left(M_k^0\left(\mathbf{x}\right) - \sigma\right)\right)}, \tag{1}$$

where $\omega$ and $\sigma$ are hyper-parameters that are respectively set to be 10 and 0.5 (following Li et al. (2018)). The final features $H_k\left(\mathbf{x}\right) \in \mathbb{R}^{E \times 1 \times 1 \times 1}$ ($k \in \{1 \ldots K\}$) are obtained through

$$H_k\left(\mathbf{x}\right) = \frac{1}{16 \times 16 \times 12} \sum_{h,w,d} G_{h,w,d}\left(\mathbf{x}\right) \cdot M_{k,h,w,d}\left(\mathbf{x}\right), \tag{2}$$

where $h$, $w$ and $d$ are height, width and depth of the corresponding features. They represent the highly-activated regions of high-level features for the respective prototype $\mathbf{p}_k$.

**Prototype and Classification Layers** The prototype layer stores a set of learned prototypes $\mathbf{p}_k \subseteq \mathbf{P}$ ($k \in \{1 \ldots K\}$), where $\mathbf{p}_k \in \mathbb{R}^{E \times 1 \times 1 \times 1}$. $K$ prototypes $\mathbf{p}_k \subseteq \mathbf{P}$ are evenly assigned to each class $c \in \{\text{HGG}, \text{LGG}\}$. The common cosine similarity is used for the calculation of the similarity score between each pair of $H_k\left(\mathbf{x}\right)$ and $\mathbf{p}_k$

$$s\left(\mathbf{x}, \mathbf{p}_k\right) = \frac{H_k\left(\mathbf{x}\right) \cdot \mathbf{p}_k}{\|H_k\left(\mathbf{x}\right)\| \|\mathbf{p}_k\|}. \tag{3}$$

Then, the similarity scores $s\left(\mathbf{x}, \mathbf{p}_k\right) \subseteq \mathbf{S}$ ($k \in \{1 \ldots K\}$) are multiplied by the weight matrix $W_{\text{cls}} \in \mathbb{R}^{K \times 2}$ in the classification layer to output the finally predicted probabilities $p\left(\mathbf{x}\right)$ for HGG and LGG (which are normalized using the softmax function).

## 2.3. Training and Loss Functions

The training of MProtoNet consists of three stages: (1) training of layers before the classification layer; (2) prototype reassignment; (3) training of the classification layer. See Appendix B for details as they are the same as those employed in previous ProtoPNet variants. The following loss functions are used during these training stages: classification loss $\mathcal{L}_{\text{cls}}$, cluster loss $\mathcal{L}_{\text{clst}}$, separation loss $\mathcal{L}_{\text{sep}}$, mapping loss $\mathcal{L}_{\text{map}}$, online-CAM loss $\mathcal{L}_{\text{OC}}$, and L1-regularization loss $\mathcal{L}_{\text{L1}}$.

Because of the imbalanced distribution of class $c \in \{\text{HGG}, \text{LGG}\}$ (HGG:LGG = 293:76), we use class weighted $\mathcal{L}_{\text{cls}}$, $\mathcal{L}_{\text{clst}}$ and $\mathcal{L}_{\text{sep}}$ from XProtoNet (Kim et al., 2021) but with a necessary transition from multi-label classification to multi-class classification. $\mathcal{L}_{\text{map}}$ originates from the occurrence loss in XProtoNet (Kim et al., 2021), while $\mathcal{L}_{\text{L1}}$ originates from the L1 regularization term of the classification layer in ProtoPNet (Chen et al., 2019).

**Online-CAM Loss** In addition to soft masking and the mapping loss, an online-CAM loss is proposed here to assist localization of the attention maps by directly utilizing image-level labels during training of the attention module. Since the last convolution operation in the mapping module of the localization layer $\text{Conv}_{\text{map}}$ is responsible for generating the attention maps, the intermediate features $I_e(\mathbf{x}) \subseteq \mathbf{I}(\mathbf{x})$ ($e \in \{1 \ldots E\}$, $I_e(\mathbf{x}) \in \mathbb{R}^{16 \times 16 \times 12}$) right before the last convolution operation are made the feature sources of online-CAM (as shown in Figure 1). To obtain a class prediction directly from $I_e(\mathbf{x})$, instead of employing average or max pooling, we choose log-sum-exp (LSE) pooling (Pinheiro and Collobert, 2015; Sun et al., 2016) which has shown improved convergence and localization performance

$$\text{LSE}\left(I_e(\mathbf{x})\right) = \frac{1}{r} \log \left[ \frac{1}{16 \times 16 \times 12} \sum_{h,w,d} \exp\left(r \cdot I_{e,h,w,d}(\mathbf{x})\right) \right], \tag{4}$$

where $h$, $w$ and $d$ are height, width and depth of the corresponding features; $r$ is a hyperparameter that is set to be 10 (following Sun et al. (2016)). Then, $\text{LSE}(\mathbf{I}(\mathbf{x})) \in \mathbb{R}^E$ is consecutively multiplied by the weight matrix $W_{\text{Conv}_{\text{map}}} \in \mathbb{R}^{E \times K}$ of $\text{Conv}_{\text{map}}$ and the weight matrix $W_{\text{cls}} \in \mathbb{R}^{K \times 2}$ in the classification layer, to obtain yet another predicted probability $p_{\text{OC}}(\mathbf{x})$ (which is different from the predicted probability $p(\mathbf{x})$ in the classification layer). $\mathcal{L}_{\text{OC}}$ is modified from $\mathcal{L}_{\text{cls}}$

$$p_{\text{OC}}(\mathbf{x}_i) = \text{Softmax}\left(\text{LSE}\left(\mathbf{I}(\mathbf{x}_i)\right) W_{\text{Conv}_{\text{map}}} W_{\text{cls}}\right),$$
$$\mathcal{L}_{\text{OC}} = \sum_i \frac{1}{N^c} \left(1 - p_{\text{OC}}^c(\mathbf{x}_i)\right)^\gamma \log p_{\text{OC}}^c(\mathbf{x}_i), \tag{5}$$

where $i$ indexes the training samples, $N^c$ denotes the total number of training samples in class $c$ and $\gamma$ is a hyper-parameter that is set to be 2 (following Kim et al. (2021)). While the online-CAM loss $\mathcal{L}_{\text{OC}}$ is useful during training, the corresponding predictions $p_{\text{OC}}(\mathbf{x}_i)$ are not performed during inference.

## 3. Experiments and Results

### 3.1. Experimental Setup

In our experiments, since the class distribution in BraTS 2020 is highly imbalanced, we use the balanced accuracy (BAC) that is the macro-average of the recall scores of the two classes HGG and LGG to evaluate the classification performance. For the interpretability metrics, we focus on correctness and localization coherence as described in Nauta et al. (2023). The correctness of a model is evaluated by the normalized area under the incremental deletion curve (Nauta et al., 2023), called the incremental deletion score (IDS). IDS measures how precisely an activation map reflects a model's decision-making process (lower is better). The

localization coherence is evaluated by the activation precision (AP) (Barnett et al., 2021). AP measures the proportion of an activation map that intersects with the human-annotated label. See Appendix C for detailed definitions of IDS and AP. To test a model's performance under different training/test splits, stratified 5-fold cross-validation is applied throughout all experiments. See Appendix D for details of the hyper-parameters during training.

There are three models compared in our experiments: CNN (with GradCAM (Selvaraju et al., 2017)), ProtoPNet (Chen et al., 2019; Barnett et al., 2021) and XProtoNet (Kim et al., 2021). For fair comparisons across models, all of them are reimplemented to share the same experimental settings as MProtoNet, except for the distinctions specified below.

**CNN (with GradCAM)**   The baseline CNN is built based on submodules of MProtoNet: the feature layer, the add-on module in the localization layer, and the classification layer. These submodules are marked as blue in Figure 1. The add-on module and the classification layer (with $W_{\mathrm{cls}} \in \mathbb{R}^{E \times 2}$) are directly connected with a global average pooling layer.

**ProtoPNet**   Compared to MProtoNet, the reimplemented ProtoPNet (adapted for 3D mpMRIs) lacks the mapping module in the localization layer and all associated loss functions. The similarity scores $s(\mathbf{x}, \mathbf{p}_k)$ are calculated by directly comparing each location in $G(\mathbf{x})$ with $\mathbf{p}_k$, followed by top-$\alpha$ average pooling (where $\alpha = 1\%$) (Barnett et al., 2021).

**XProtoNet**   MProtoNet is actually the reimplemented XProtoNet (adapted for 3D mpMRIs) with the addition of soft masking and the online-CAM loss.

### 3.2. Results

Table 1 shows all quantitative results from the 5-fold cross-validation experiments, including three versions of MProtoNet (A: without online-CAM loss, B: without soft masking, C: complete version). The results represent the contributions of all modalities since the fusion of modalities occurs at the very beginning. Given that MProtoNet is most similar to XProtoNet, 5-fold cross-validated paired Student's t-tests (null hypothesis: paired results have identical means) against XProtoNet are performed on all results for MProtoNet A/B/C.

Comparing classification performance (BAC), there are no statistically significant differences among all models, most notably between MProtoNet A/B/C and XProtoNet. This is desirable as the same 3D ResNet backbone is shared across all models and appears to be well-suited for this task. The introduction of the interpretable components in our MProtoNet does not necessarily improve or hurt the classification performance. However, one of the most valuable outcomes is that it can provide case-based prototypical explanations during inference, as shown in Figure 2.

The complete version MProtoNet C achieves the best performance in terms of both correctness (IDS) and localization coherence (AP), excelling in interpretability. Specifically, both soft masking and the online-CAM loss are very important for statistically significant improvements in IDS ($p = 0.031$) and AP ($p < 0.001$) over XProtoNet, resulting in the best IDS of $0.079 \pm 0.034$ and the best AP of $0.713 \pm 0.058$. Since we do not use any annotation information during training, the poor performance of the original ProtoPNet (even with top-$\alpha$ average pooling) is expected as previously shown in Barnett et al. (2021). Figure 3 clearly shows the improvement of MProtoNet C over all other models in terms of localization coherence. See Appendix E for more visualization examples.

Table 1: A summary of all results in the form of [mean ± standard deviation] from 5-fold cross-validation. Bold indicates the best performance. For MProtoNet A/B/C, the $p$-value from the paired t-test against XProtoNet is shown under each result. Keys: AM – attention map, SM – soft masking, OC – online-CAM, BAC – balanced accuracy, IDS – incremental deletion score, AP – activation precision.

| Model | Condition | | | Classification | Interpretability | |
|---|---|---|---|---|---|---|
| | AM | SM | OC | BAC | IDS | AP |
| CNN (with GradCAM) | | | | $0.865 \pm 0.026$ | $0.112 \pm 0.049$ | $0.099 \pm 0.030$ |
| ProtoPNet | | | | $0.868 \pm 0.032$ | $0.609 \pm 0.164$ | $0.007 \pm 0.001$ |
| XProtoNet | ✓ | | | $\mathbf{0.870 \pm 0.021}$ | $0.170 \pm 0.041$ | $0.203 \pm 0.030$ |
| MProtoNet A | ✓ | ✓ | | $0.868 \pm 0.050$ ($p=0.929$) | $0.150 \pm 0.088$ ($p=0.647$) | $0.568 \pm 0.125$ ($p=0.004$) |
| MProtoNet B | ✓ | | ✓ | $0.865 \pm 0.015$ ($p=0.360$) | $0.103 \pm 0.020$ ($p=0.069$) | $0.204 \pm 0.028$ ($p=0.963$) |
| MProtoNet C | ✓ | ✓ | ✓ | $0.858 \pm 0.048$ ($p=0.516$) | $\mathbf{0.079 \pm 0.034}$ ($p=0.031$) | $\mathbf{0.713 \pm 0.058}$ ($p<0.001$) |

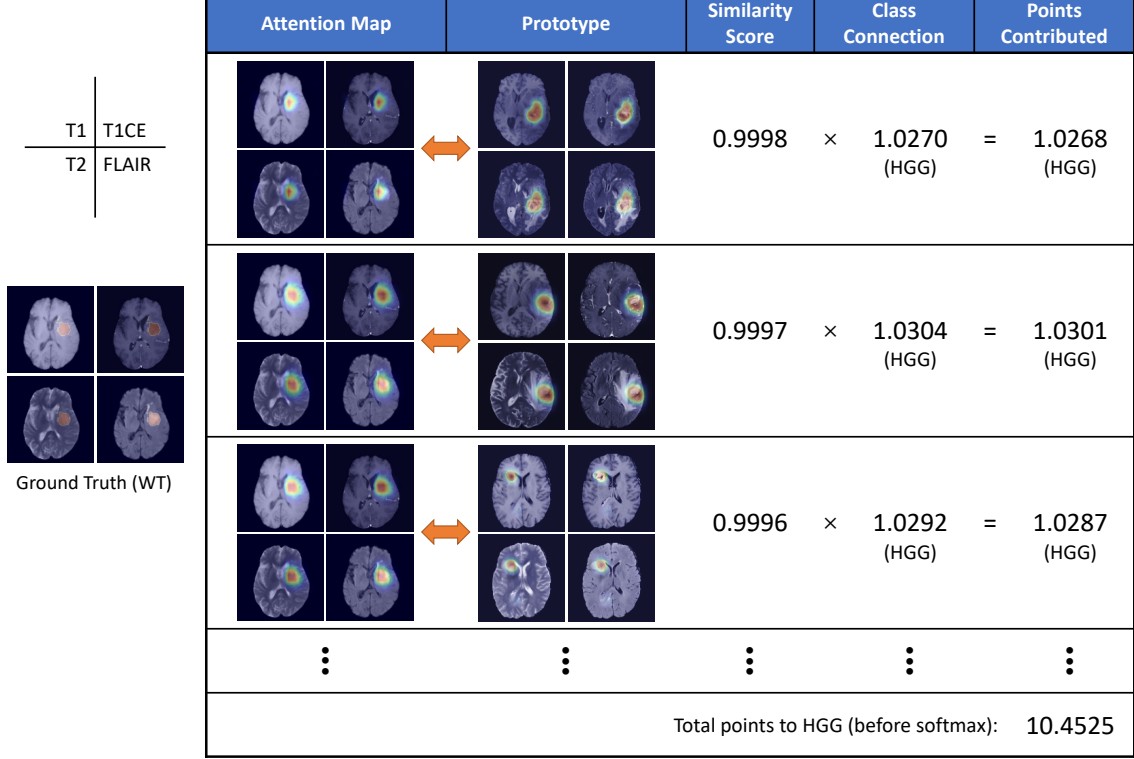

Figure 2: Demonstration of the case-based reasoning in MProtoNet. It can provide case-based prototypical explanations with both attention maps and similarity scores.

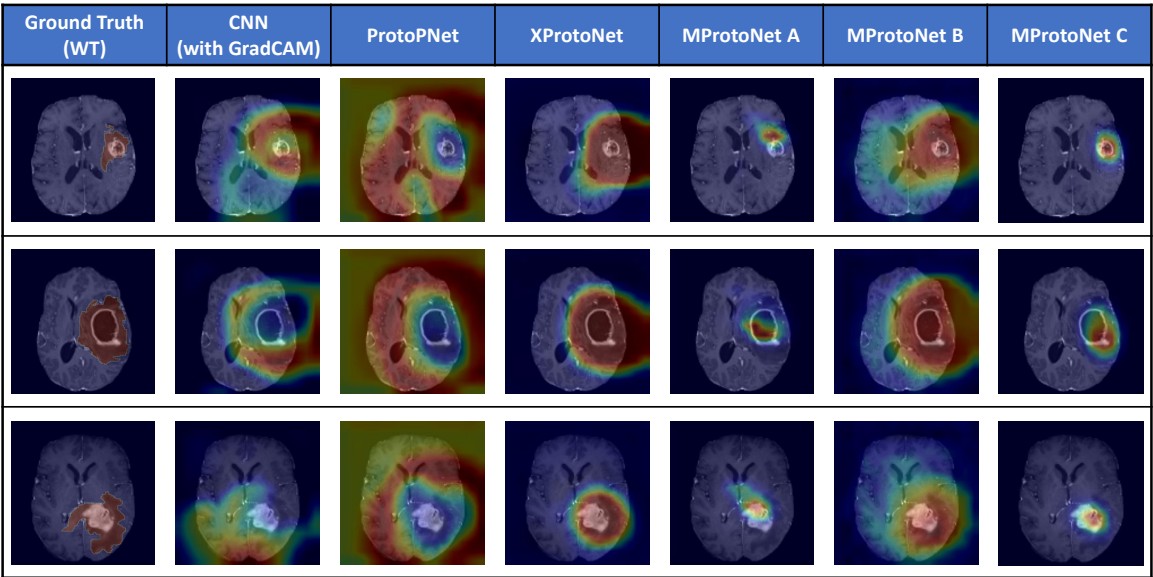

Figure 3: Visualization examples of the localization coherence results from different methods (shown with the T1CE modality). MProtoNet C clearly outperforms all other models, showing the importance of both soft masking and the online-CAM loss.

## 4. Conclusions and Future Work

In this work, we propose MProtoNet to apply case-based interpretable deep learning to brain tumor classification with 3D mpMRIs, targeting a goal of precisely localizing attention regions. The proposed new attention module with soft masking and online-CAM loss helps MProtoNet achieve the best performance on interpretability in terms of both correctness and localization coherence. Meanwhile, the classification accuracy is on par with baseline CNN and other ProtoPNet variants specifically designed for medical images. No annotation information is required during training, showing the great potential of MProtoNet on 3D medical image applications wherein fine-grained annotation labels are difficult to obtain. The ability to simultaneously output attribution maps and case-based prototypical explanations also makes MProtoNet a potential alternative to current 3D CNNs in many 3D medical imaging tasks.

There are several potential aspects for future improvements over MProtoNet. Firstly, rather than fixed assignments of prototypes before and after training, a dynamic or even shared assignment (Rymarczyk et al., 2021, 2022) might be more suitable for some medical image applications. Secondly, instead of fusing the multiple modalities at the beginning, we shall also test fusion at the end to analyze each individual modality like some gradient-based methods (Jin et al., 2022, 2023). Lastly, potential combinations with other interpretable methods such as concept-based models (Koh et al., 2020) might further improve MProtoNet's interpretability on much more complicated medical image applications.

## Acknowledgments

This study was supported by the National Natural Science Foundation of China (62071210); the Shenzhen Science and Technology Program (RCYX20210609103056042); the Shenzhen Science and Technology Innovation Committee (KCXFZ2020122117340001); the Shenzhen Basic Research Program (JCYJ20200925153847004, JCYJ20190809120205578); the Natural Sciences and Engineering Research Council of Canada.

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

## Appendix A. Preprocessing and Data Augmentation

### A.1. Preprocessing

All mpMRI data in BraTS 2020 go through a standardized preprocessing pipeline including co-registration to a common anatomical template (SRI24 (Rohlfing et al., 2010)), resampling to a $1{\times}1{\times}1$ mm$^3$ resolution, and skull-stripping. After original preprocessing, each image has a size of $240{\times}240{\times}155$ voxels ($1{\times}1{\times}1$ mm$^3$ resolution). The images are cropped around the borders to $192{\times}192{\times}144$ voxels, and then further down-sampled to $128{\times}128{\times}96$ voxels ($1.5{\times}1.5{\times}1.5$ mm$^3$ resolution) to enable efficient training on GPUs. After that, intensity normalization is performed on each mpMRI image of a specific modality by subtracting the individual mean and dividing by the individual standard deviation.

### A.2. Data Augmentation

During training, we follow the online data augmentation pipeline of nnU-Net (Isensee et al., 2021), which has already been successfully applied to many 3D medical imaging tasks, to overcome the limited size issue of the training data. The augmentations are applied stochastically in the following sequence: (1) rotation and scaling; (2) Gaussian noise; (3) Gaussian blur; (4) brightness augmentation; (5) contrast augmentation; (6) simulation of low resolution; (7) gamma augmentation; (8) mirroring. Detailed parameters for each augmentation operation can be found in Isensee et al. (2021).

## Appendix B. Training Stages

**Stage-1: Training of Layers Before the Classification Layer**    During the first training stage, we train all layers except the classification layer to obtain a meaningful embedding space for the prototypes. The overall loss function is

$$\mathcal{L}_{\text{stage-1}} = \lambda_{\text{cls}}\mathcal{L}_{\text{cls}} + \lambda_{\text{clst}}\mathcal{L}_{\text{clst}} + \lambda_{\text{sep}}\mathcal{L}_{\text{sep}} + \lambda_{\text{map}}\mathcal{L}_{\text{map}} + \lambda_{\text{OC}}\mathcal{L}_{\text{OC}}, \tag{6}$$

where $\lambda_{\text{cls}}$, $\lambda_{\text{clst}}$, $\lambda_{\text{sep}}$, $\lambda_{\text{map}}$ and $\lambda_{\text{OC}}$ are coefficients of respective losses.

**Stage-2: Prototype Reassignment**    To achieve the goal of case-based reasoning, i.e., to be able to view each prototype as a specific patch of a training sample, we need to replace the representation of each prototype $\mathbf{p}_k \subseteq \mathbf{P}$ ($k \in \{1 \ldots K\}$) with the nearest processed feature $H_k(\mathbf{x})$ (without any online data augmentation for $\mathbf{x}$) after several iterations in stage-1. Therefore, we can treat the corresponding training sample $\mathbf{x}$ with the attention map $M_k(\mathbf{x})$ superimposed on it as the visualization of prototype $\mathbf{p}_k$.

**Stage-3: Training of the Classification Layer**    After reassigning prototypes in stage-2, we train the classification layer separately to learn a weight matrix $W_{\text{cls}} \in \mathbb{R}^{K \times 2}$ that represents the contributions of the prototypes for classification without changing their representation. The overall loss function is

$$\mathcal{L}_{\text{stage-3}} = \lambda_{\text{cls}}\mathcal{L}_{\text{cls}} + \lambda_{\text{L1}}\mathcal{L}_{\text{L1}}, \tag{7}$$

where $\lambda_{\text{cls}}$ and $\lambda_{\text{L1}}$ are coefficients of respective losses.

## Appendix C. Evaluation Metrics for Interpretability

### C.1. Correctness: Incremental Deletion Score (IDS)

Incremental deletion (Nauta et al., 2023) is a commonly adopted method to assess the interpretability of an activation map by incrementally deleting features (e.g., replaced with 0) in the input according to the activation map values (from high values to low values). The top-valued features are deleted at the beginning. If the curve of classification metric drops faster, the activation map more precisely reflects a model's decision-making process. To quantify the sharpness of the curve of incremental deletion, incremental deletion score (IDS) is defined here as the normalized area under the curve and within the bounds of start and end, as shown in Figure 4. The lower the IDS, the better performance in terms of correctness.

### C.2. Localization Coherence: Activation Precision (AP)

The localization coherence is evaluated by the activation precision (AP) (Barnett et al., 2021)

$$\text{AP} = \frac{|H(\mathbf{x}) \cap T(\text{UpSample}(M(\mathbf{x})))|}{|T(\text{UpSample}(M(\mathbf{x})))|}, \tag{8}$$

where $H(\mathbf{x})$ is the human-annotated label for the WT region of $\mathbf{x}$, $M(\mathbf{x})$ is the activation map for $\mathbf{x}$ and $T(\cdot)$ is a threshold function. In this work, $T(\cdot)$ is set to be a binary function

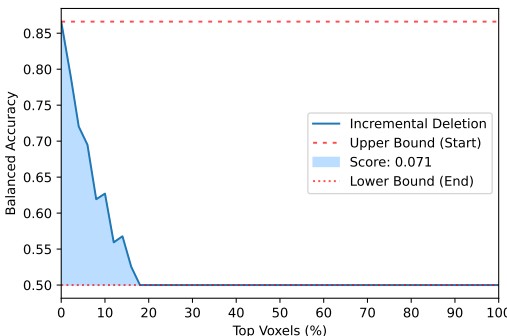 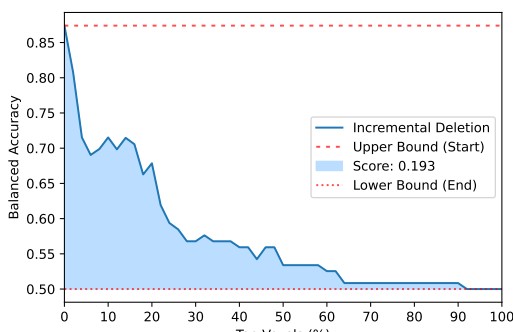

Figure 4: Demonstration of the incremental deletion score (IDS).

with the threshold 0.5. For MProtoNet as well as XProtoNet

$$M\left(\mathbf{x}\right) = \frac{1}{K^c} \sum M_k^c\left(\mathbf{x}\right), \tag{9}$$

where $c$ denotes the correct class of $\mathbf{x}$ and $K^c$ denotes the total number of prototypes assigned to class $c$. For CNN (with GradCAM), GradCAM is used to generate the activation map $M\left(\mathbf{x}\right)$ from the last convolution layer of the CNN after training. For ProtoPNet, the average of the intermediate results right before top-$\alpha$ average pooling is used as the activation map $M\left(\mathbf{x}\right)$ (following Barnett et al. (2021)).

## Appendix D. Hyper-parameters During Training

Following some latest practices from Liu et al. (2022), MProtoNet is trained by the AdamW optimizer with a baseline learning rate of 0.001 and a weight decay coefficient of 0.01. During each cross-validation iteration, MProtoNet is trained for 100 epochs (stage-1) with a batch size of 32 and a specific learning rate scheduler (linear warm-up for the first 20 epochs and cosine annealing for the remaining 80 epochs). After every 10 epochs of stage-1, there is a step of stage-2 (with only prototype reassignment) and 10 epochs of stage-3 (where MProtoNet is trained by the Adam optimizer with a constant learning rate of 0.001). The number of embedding channels $E$ is set to be 128 and the number of prototypes $K$ is set to be 30 (15 for each class). The coefficients of loss functions are set as follows: $\lambda_{\text{cls}}$ as 1, $\lambda_{\text{clst}}$ as 0.8, $\lambda_{\text{sep}}$ as 0.08, $\lambda_{\text{map}}$ as 0.5, $\lambda_{\text{OC}}$ as 0.05 and $\lambda_{\text{L1}}$ as 0.01.

## Appendix E. More Visualization Examples of Localization Coherence

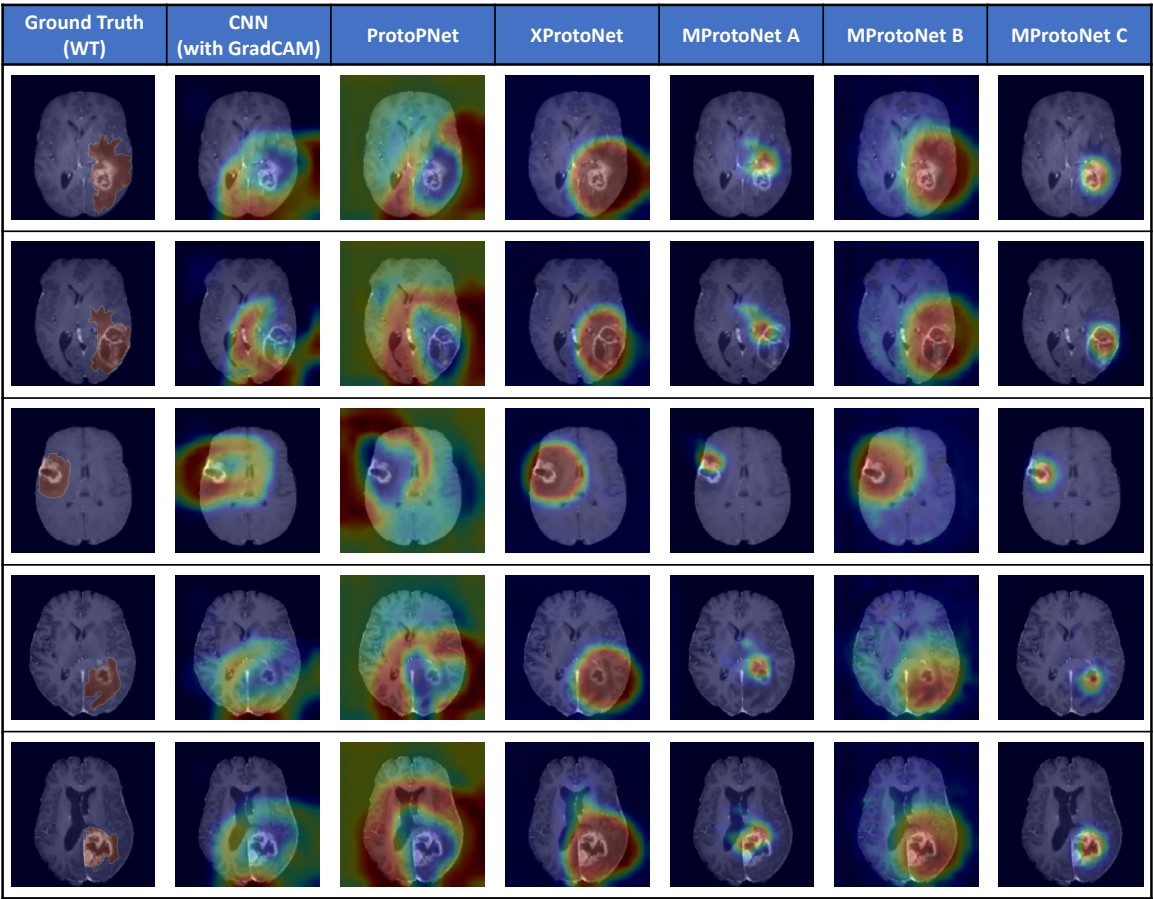

Figure 5: More visualization examples of the localization coherence results from different methods (shown with the T1CE modality).

