# OpenReview forum: "MProtoNet: A Case-Based Interpretable Model for Brain Tumor Classification with 3D Multi-parametric Magnetic Resonance Imaging"
_MIDL.io/2023/Conference — MIDL 2023 Oral_

### Official Review · Reviewer_g47b · 2023-01-26

**Confidence:** 4
**Preliminary Rating:** 4
**Recommendation:** Oral

**Summary:**

This manuscript discusses the use of deep convolutional neural networks (CNNs) in medical imaging and the challenges of interpretability. The authors propose a new model called MProtoNet, which is an extension of ProtoPNet for use in multi-modal 3D medical images, specifically brain tumor classification with 3D multi-parametric magnetic resonance imaging (mpMRI) data. MProtoNet includes a new attention module with soft masking and online-CAM loss that helps to sharpen attention maps and directly utilizes image-level labels during training. The results show that MProtoNet achieves statistically significant improvements in interpretability metrics compared to baseline CNNs and other ProtoPNet variants.

**Strengths:**

-MProtoNet achieves the best performance in terms of both correctness and localization coherence, excelling in interpretability.

-The attention module with soft masking and online CAM loss is a novel solution to the differences between 2D natural images and 3D mpMRIs.

-Results show that MProtoNet achieves statistically significant improvements in interpretability metrics of both correctness and localization coherence without any annotated labels during training when compared with baseline CNN and several ProtoPNet variants.


**Weaknesses:**

-It is not clear from the abstract how well MProtoNet performs in comparison to other existing approaches for interpretability in medical imaging.

-The authors should provide a few more examples.

-As mentioned by the authors, it's possible that incorporating other interpretable methods, such as concept-based models, could enhance MProtoNet's interpretability in more complex medical image applications.


**Deanonymize Review:**

yes

**Paper Type:**

both

**Questions To Address In The Rebuttal:**

It would be useful to see more details on how well the approach performs in comparison to other existing approaches for interpretability in medical imaging. The proposed approach has great potential for other 3D medical image applications, it would be interesting to see how well the proposed approach can be extended to other medical imaging tasks.

1. Please provide more information about how MProtoNet compares to other state-of-the-art models for 3D medical imaging?
2. Further, can you provide more information about the images shown in Appendix E?
3. Additionally, please explain why some of the models are generating maps beyond the boundary of the skull (appendix E)?

---

### Official Review · Reviewer_NoMC · 2023-02-04

**Confidence:** 4
**Preliminary Rating:** 5
**Recommendation:** Poster

**Summary:**

The authors present a case-based interpretable model for medical applications and evaluate their proposed method on brain tumor classification (high grade vs. low grade). The method is based on ProtoNet. The evaluation shows a statistically significant superior classification performance and improved interpretability.

**Strengths:**

The paper is clearly motivated and discusses prior work, research gaps, and specific contributions. The writing is clear, and the figures and tables help the reader to understand the experiments and the results. The experimental setup is described clearly and includes an ablation study and a reasonable baseline.

**Weaknesses:**

The authors use a five-fold cross-validation for each method they compare. The fine-tuning and parameter setting (and potentially the loss-weighting?) should, in my opinion, be limited to a single split of this five-fold CV, which seems not to be the case.

**Deanonymize Review:**

no

**Detailed Comments:**

I appreciate the very detailed experimental setup description, making the work reproducible. The appendix provides additional information to also visually appreciate the localization coherence of the different models.

**Paper Type:**

methodological development

**Questions To Address In The Rebuttal:**

If you would apply your method to diseases with a less localized presentation than a tumor (e.g., MRI of Alzheimer's patients with distributed atrophy patterns): Would you need to, for instance, need to reduce the online CAM loss weight?

---

### Official Review · Reviewer_5cxA · 2023-02-06

**Confidence:** 4
**Preliminary Rating:** 3

**Summary:**

The paper proposes a case-based interpretable deep learning model for brain tumor classification with 3D multi-parametric magnetic resonance imaging (mpMRI) data. The model is called MProtoNet and is an extension of ProtoPNet, which is a case-based reasoning model for image classification. More specifically, the authors added an online-CAM loss and a soft masking to XProtoNet, and replaced 2D filters with 3D. Overall, the paper is of interest, but some questions need to be addressed.

**Strengths:**

- Interpretable models in medical imaging are an ongoing research topic and of high interest.
- Use of soft masking to refine attention maps from XProtoNet
- Online-CAM loss function that uses image-level labels during image analysis.

**Weaknesses:**

- I find the paper unfocused on the main contribution and rather re-explaining what was already done in XProtoNet.
- The paper does not mention the computational cost of the proposed method and how it compares to other state-of-the-art methods in terms of speed and memory usage.

**Deanonymize Review:**

no

**Detailed Comments:**

- I suggest that the authors re-write the paper putting the main contribution forward, and refer to XProtoNet when needed. Avoid using sentences such as "L1 was introduced, we introduce focal loss"...etc, these were introduced in XProtoNet and may be confusing to the reader.
- I suggest to re-write the sentences such as " Given that XProtoNet is most similar to MProtoNet" to "Given that MProtoNet is most similar to XProtoNet " ... throughout the whole paper. XProtoNet is the original architecture and MProtoNet is a variant with an online-CAM loss.
- I suggest to focus on the differences between MProtoNet and XProtoNet, for the rest of the description.
- Please move the appendix to the main text, this part is the main work that has been done  and important to the reader.
- Related  work could be extended. Several papers for interpretable deep learning models using class-label to guide attention were proposed for natural images and medical imaging, for instance:
1- Zhou, Y., Zhu, Y., Ye, Q., Qiu, Q. and Jiao, J., 2018. Weakly supervised instance segmentation using class peak response. In Proceedings of the IEEE conference on computer vision and pattern recognition (pp. 3791-3800).
2- Amyar, A., Modzelewski, R., Vera, P., Morard, V. and Ruan, S., 2022. Weakly Supervised Tumor Detection in PET Using Class Response for Treatment Outcome Prediction. Journal of Imaging, 8(5), p.130.


**Paper Type:**

methodological development

**Questions To Address In The Rebuttal:**

- Please see above questions.
- Please include quantitative results in the abstract.
- "MProtoNet achieves statistically significant improvements" Please provide statistical analysis to show that it is "statistically" different.
- "The proposed MProtoNet can be easily extended to other 3D medical image applications". Please do not make assumptions without any results. The paper does not show the generazibility of the method to other 3D medical image applications. Please focus the conclusion on the work that has been done only.
- "The source code will be made publicly accessible upon acceptance." I would highly encourage the authors to make their code available during review.
- Please add a take away message to the figures.

---

### Meta-Review · Area_Chair_s1Yc · 2023-02-24

**Recommendation:** Accept (Poster)
**Confidence:** 4

**Metareview:**

The paper addresses the important topic of interpretability and the approach proposed is applicable beyond brain tumors. The paper is clearly written, with good explanation of improvements over the previous method (ProtoPNet and XProtoNet) that this builds on. The performance is good compared to previous work, but only evaluated on a tumor application.

Pros:
- Hot topic and has impactful applications
- Open data is used
- Contributions clearly laid out with respect to ProtoPNet and XProtoNet (in response to reviewer comment)
- Experiments clearly described
- Good experimental performance

Cons:
- Source code not available for review, just promised for later
- Evaluation limited to single disease - low-hanging fruit to test on other diseases
- While it is good to perform hypothesis testing on performance metrics when comparing models, the choice of a paired student t-test is odd: this is meant for interval-scale, normally distributed data, but i'm not sure IDS or AP satisfy these - AP is a proportion (so maybe use Wald test or likelihood ratio test ), and IDS is an area under curve so a permutation test could be used instead (see e.g. Venkatraman & Begg, Biometrika, 1996).

---

### Meta-Review · Program_Chairs · 2023-02-28

**Recommendation:** Accept (Oral)
**Confidence:** 4

**Metareview:**

This an excellent and well-written study, with interest to the community. It addresses an important problem of interpretablity in medical imaging. I recommend it for an oral presentation.